# SelfDreamer: Dual-Prototypical Regularization for Frame-Masked Model-Based Reinforcement Learning

## Abstract

In the realm of reinforcement learning (RL), the conventional approach involves training agents in unknown environments using extensive experiences comprising high-dimensional state representations (typically images), actions, and rewards. However, this standard setup imposes substantial data transmission overhead in scenarios where edge devices are employed for data collection, and cloud servers are utilized for model training. This paper introduces a novel paradigm termed "frame-masked RL," which is devised to enhance data efficiency while examining the impact on existing methods. Concurrently, we introduce a model-based algorithm, "SelfDreamer," tailored to mitigate the information loss incurred due to frame masking. SelfDreamer leverages action-transition dual prototypes to embed action information within the world model and align the hidden states in the representation space. Empirical evaluations reveal that SelfDreamer consistently outperforms state-of-the-art methods across six continuous control tasks sourced from the DeepMind Control Suite, demonstrating superior or comparable performance while utilizing only half of the observations from the environment.

## 1 Introduction

Reinforcement learning (RL) is a foundational paradigm within machine learning, primarily dedicated to the training of autonomous agents for effective decision-making and continuous control tasks. In this context, RL agents typically operate in environments characterized by a degree of uncertainty, wherein they receive information regarding the current states and associated rewards iteratively. Recent advancements in RL have witnessed a predilection for representing state signals in the form of images. However, this prevalent approach may raise concerns, especially in cloud computing scenarios where training data collection and model learning occur at disparate endpoints. Notably, when gathering trajectories from resource-constrained edge devices, such as drones or robotic arms, and training a consolidated policy through a central server, the storage and transmission overhead incurred by image-based state representations become substantial. Consequently, the exigency to address the challenges posed by image-based reinforcement learning under conditions of sparse state signals, wherein the performance must be preserved despite concealing a portion of the training data, becomes evident (Fig. 1a).

Model-based reinforcement learning (MBRL), as originally conceived by Sutton (1991), emerges as a promising candidate to address the aforementioned challenge. MBRL agents, although devoid of direct interaction with the physical environment, acquire knowledge from a latent world model that simulates real-world dynamics, thereby enhancing data efficiency. In the aforementioned context, the viability of the world model hinges on its capacity to glean meaningful insights from limited state images, while simultaneously having unrestricted access to lightweight scalar representations of actions and rewards. If such a world model can be successfully acquired and accurately emulates the true environment, it offers an enticing solution to the edge-cloud co-design predicament.

Within the domain of image-based MBRL, the Dreamer framework, as introduced by Hafner et al. (2020), stands out for its commendable performance. Dreamer represents a significant milestone as the first MBRL agent to outperform established model-free RL agents, as delineated by Barth-Maron et al. (2018); Hessel et al. (2018), demonstrating superior sample efficiency in both discrete

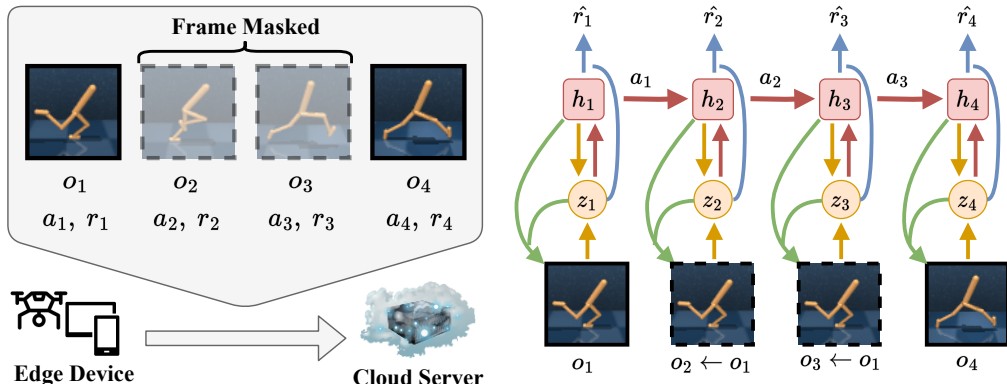

(a) Frame masking for edge-cloud co-design.  (b) Model-based RL under sparse state signals.

Figure 1: (a) Illustration of the system configuration featuring a central server and multiple edge devices, where the RL model is trained using data collected from resource-constrained edge devices by a resource-abundant server. To mitigate power and storage overhead on small-scale devices, the concept of frame masking is considered, wherein only observations at select timesteps are stored and transmitted, while lower-dimensional action and reward vectors or scalars are retained. (b) Depiction of the training process within Dreamer , a state-of-the-art MBRL framework , incorporating frame masking. The world model is trained through three objectives: reward prediction, transition prediction, and observation-related tasks (represented by blue, orange, and green arrows). In this study, we pad the masked frames with the previous valid frames, e.g., $o2$ and $o3$ are padded by $o1$, to investigate the potential degradation of existing methods and strategies to restore their performance.

and continuous control domains. Dreamer's world model operates by encoding high-dimensional visual data into a compact latent space, thereby enhancing computational efficiency. This low-dimensional state space, furnished by the world model, facilitates policy training through gradient-based algorithms integrated into a differentiable architecture.

The role of image state signals in the context of MBRL is pivotal, as direct reward maximization often falters due to the inherent sparsity and noise of rewards. Dreamer addresses this challenge by employing a reconstruction loss on sequences of visual observations, effectively framing it as an auxiliary task that bridges the gap between the model and the real-world environment. Concurrently, other studies, such as those by Nguyen et al. (2021); Deng et al. (2022), have proposed alternative strategies to bolster the robustness of the latent space, eschewing the reliance on reconstruction and achieving enhanced performance in regular continuous control tasks.

Despite the commendable achievements of contemporary MBRL, a notable challenge persists concerning the acquisition of a reliable world model from partial training data, which stems from the reliance on visual observations. Notably, the simulated world model may exhibit distributional disparities compared to the actual environment, leading to unforeseen performance deviations in downstream policies. Even attempts to mitigate this issue, such as padding missing frames with the latest valid frames (Fig. 1b), often result in a flattened latent space, undermining subsequent reward prediction and policy learning.

This paper delves into the intricacies of MBRL when confronted with incomplete state signals, specifically in the form of visual observations. Additionally, it explores potential solutions to ameliorate the performance degradation attributed to information sparsity. Drawing inspiration from prototypical learning in computer vision (Snell et al., 2017) and prior work in MBRL (Deng et al., 2022), we propose an innovative algorithm named "SelfDreamer." This algorithm capitalizes on the concept of action-transition dual-prototypical learning, introducing a self-supervised regularization mechanism that enforces consistent transitions for similar actions. This regularization aids in conferring consistency and alignment to the latent space, particularly in the context of image-missing states. Subsequently, we evaluate the efficacy of the proposed algorithm using the standard DeepMind Control Suite, applying frame masking to a subset of images from the environments. The empirical results demonstrate that SelfDreamer consistently outperforms three state-of-the-art MBRL

methods across six continuous control tasks, achieving higher final returns and, notably, superior or equivalent performances while utilizing only half of the state images.

The contributions of this work can be succinctly summarized as follows:

- This study represents a pioneering exploration of sparse state signals in reinforcement learning, frame-masked RL, showcasing a significant enhancement in data efficiency. The results shed light on a new line of research.
- In addition to outlining this novel research direction, we introduce SelfDreamer, which incorporates a dual-prototypical mechanism featuring action-consistent transitions to embed action information into the MBRL world model and reform the representation space.
- Extensive experimental evaluations affirm the empirical effectiveness of SelfDreamer, as it consistently outperforms state-of-the-art RL methods under standard settings and delivers superior or comparable policies while achieving double data efficiency in frame-masked RL scenarios.

## 2 PRELIMINARIES

### 2.1 FRAME-MASKED REINFORCEMENT LEARNING

---

**Algorithm 1:** Edge-cloud co-design with frame-masking

---

Initialize an empty cloud dataset $\mathcal{D} = \{\}$.
Initialize policy parameters $\phi$.
**while** *not converged* **do**

    /\* Experience Collecting by the Edge Devices    \*/
    Receive $\phi$ from the server.
    $o_1 = env.reset()$
    **for** *timestep $t = 1..T$* **do**

        // Mask frames periodically.
        **if** $t \mod P \neq 1$ **then**
            $o_t \leftarrow o_{t-1}$
        $a_t \sim \pi_\phi(a_t|o_t)$
        $r_t, o_{t+1} \leftarrow env.step(a_t)$

    Transmit experiences to the global dataset $\mathcal{D} \leftarrow \mathcal{D} \cup \{(o_t, a_t, r_t)\}_{t=1}^{T}$.
    /\* Model Training by the Cloud Server    \*/
    **for** *update step $c = 1..C$* **do**

        Sample $B$ training sequences $\{(o_t, a_t, r_t)\}_{t=k}^{k+L} \sim \mathcal{D}$.
        Update $\phi$ by arbitrary RL algorithm.

**Notations**

| | |
|---|---|
| $o_t$ | observation at time step $t$ |
| $a_t$ | action at time step $t$ |
| $r_t$ | reward at time step $t$ |
| $\pi_\phi(a_t|o_t)$ | policy with parameters $\phi$ |

**Hyperparameters**

| | |
|---|---|
| $T$ | length of interaction |
| $P$ | frame-masking period |
| $C$ | collect interval |
| $B$ | batch size |
| $L$ | sequence length |

**Frame-masking**

For data compression, masked frames are dismissed during experience transmission and are recovered by repeating the previous unmasked frames at the server end.

---

Reinforcement learning is conventionally formulated within the framework of Markov Decision Processes (MDP; Sutton (1991)), characterized by a state space $\mathcal{S}$, an action space $\mathcal{A}$, a reward function $\mathcal{R}$, and a transition function $\mathcal{T}$, sometimes accompanied by a discount factor $\gamma$. The fundamental objective of an MDP agent is to maximize the cumulative reward by engaging with an unknown environment, where each interaction at time step $t$ is typically represented as tuples $(s_t, a_t, r_t, s_{t+1})$. These tuples signify that (1) the agent receives the current state from the environment and responds with a valid action, and (2) the environment executes the specified action, returns the associated reward, and provides the subsequent state.

However, in the realm of most RL research, direct access to the internal states of the environment is typically unavailable, with only observations being accessible, often manifesting as visual perceptions (hence, we denote $o_t$ instead of $s_t$ as the input to the policy in this paper). The extensive trial-and-error nature of RL training processes can render high-dimensional state signals impractical for

cloud computing frameworks reliant on data collected from resource-constrained edge devices (Dai et al., 2022). For instance, in standard benchmarks like the DeepMind Control Suite (Tassa et al., 2018) and the Atari Benchmark (Bellemare et al., 2013), agents are required to process large numbers of 64x64 and 210x160 color images, amounting to $500K$ and $50M$ observations, respectively.

In response to these challenges, we introduce a novel paradigm termed "frame-masked RL," designed to reduce the demand for environment frames during both training and testing, compared to the conventional RL setting. As outlined in Algorithm 1, edge devices are deployed to interact with the environment and collect experiences $\{(o_t, a_t, r_t)\}_{t=1}^T$. However, observations are sampled only at intervals of $P$ time steps (referred to as the "frame-masking period"), with the masked frames padded using the most recent valid observations. This approach maintains the integrity of the entire trajectory by interspersing genuine frames as anchors. Consequently, the agent operates under conditions of sparse state signals, leading to a reduction in data storage and transmission overhead by a factor of $P$ (with the size of action and reward signals being relatively negligible). Importantly, rewards are still computed within the original state space, i.e., the true observation is discarded after the reward function calculation, preserving valuable information. Simultaneously, a cloud server is employed to receive trajectories from the edge device, and these can be efficiently compressed and reconstructed due to the periodic repetition of observations. Subsequently, the desired model can be trained within the server using any RL algorithm of choice and then returned to the edge device for further iterations. It is worth noting that frame skipping (Mnih et al., 2013) also involves disregarding responses from the environment, but it accomplishes this by repeating actions to capture sufficient dynamics, especially in high-frequency games.

## 2.2 MODEL-BASED REINFORCEMENT LEARNING

In the context of this paper, our primary focus lies in model-based reinforcement learning with frame masking. As such, we first introduce the state-of-the-art framework, Dreamer (Hafner et al., 2020). While we will briefly touch on the training of the policy, it is essential to note that this aspect is not the central emphasis of our work.

**World-model learning.** The central objective of Dreamer is to create a compact world model that encapsulates the transition and reward structures. This world model takes the form of a recurrent state-space model (RSSM; Hafner et al. (2019)).

$$
\begin{aligned}
&\text{Recurrent model:} && h_t = f_\theta(h_{t-1}, z_{t-1}, a_{t-1}) \\
&\text{Representation model:} && z_t \sim p_\theta(z_t \mid h_t, o_t) \\
&\text{Transition predictor:} && \hat{z}_t \sim q_\theta(\hat{z}_t \mid h_t) \\
&\text{Reward predictor:} && \hat{r}_t \sim q_\theta(\hat{r}_t \mid h_t, z_t) \\
&\text{Observation predictor:} && \hat{o}_t \sim q_\theta(\hat{o}_t \mid h_t, z_t)
\end{aligned}
\tag{1}
$$

For a comprehensive illustration, refer to Fig. 1b, which depicts the following key components: (1) The recurrent model (red arrows), which comprises a GRU (Cho et al., 2014) responsible for encoding previous actions and observations into the deterministic latent variable $h_t$. (2) The representation model and the transition predictor (yellow arrows), which introduce the stochastic stream $z_t$ through variational encoding. (3) The reward predictor (blue arrows), which facilitates downstream policy learning by emulating the reward function. (4) The observation predictor (green arrows), which contributes to model coherence by reconstructing observations from the model states. These components are jointly trained by maximizing the evidence lower bound (ELBO; Jordan et al. (1999)).

$$
\sum_{t=1}^T \mathbb{E}_q \left[ -\underbrace{D_{KL}(p(z_t \mid h_t, o_t) \| q(\hat{z}_t \mid h_t))}_{\mathcal{J}_{KL}^t} + \underbrace{\log q(r_t \mid h_t, z_t)}_{\mathcal{J}_R^t} + \underbrace{\log q(o_t \mid h_t, z_t)}_{\mathcal{J}_O^t} \right]
\tag{2}
$$

A descendant of Dreamer, known as DreamerPro (Deng et al., 2022), serves as the basis for our method. DreamerPro augments Dreamer by replacing the reconstruction loss $\mathcal{J}_O^t$ with two cluster assignment tasks. This modification enhances the robustness of the RSSM by aligning the model's capacity with the underlying nature of states, rather than attempting to fit noisy observations.

**Policy learning by world simulation.** Dreamer operates by alternating between the training of the world model and policy. Both an actor and a critic, consisting of MLPs with ELU activations, are employed to learn from the latent trajectories generated by the world model. The simulation process commences at each non-terminal state $s_t = [h_t, z_t]$ encountered during world model learning. At each step of imagination, an action $a_\tau$ is sampled from the actor's stochastic policy. Predicted rewards $\hat{r_\tau}$ and subsequent states $s_{\hat{\tau+1}}$ are generated based on the learned world model. Utilizing these simulated trajectories, the actor refines its policy using biased but low-variance straight-through gradients (Kingma & Welling, 2013) and explores by regularizing the output entropy. Simultaneously, the critic is trained to approximate the $\lambda$-return (Schulman et al., 2016) using a squared loss.

## 3 SELFDREAMER

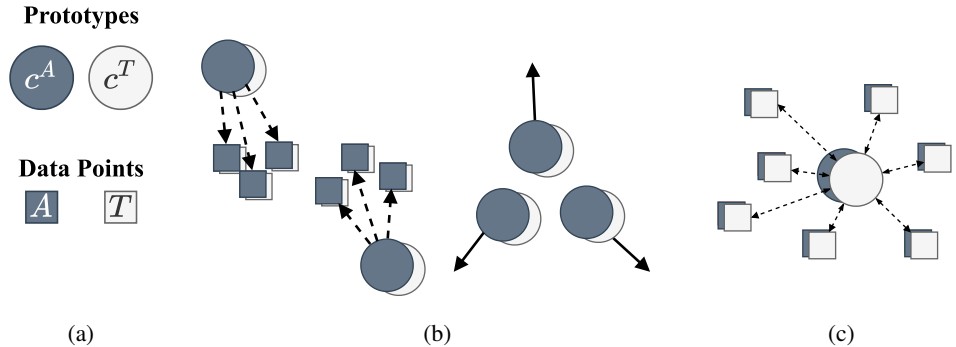

(a)         (b)         (c)

Figure 2: Illustration of SelfDreamer. (a) SelfDreamer leverages action and transition coupling to learn pairs of prototypes, comprising an action prototype $c^A$ and a transition prototype $c^T$. (b) The action prototypes are learned by assigning data pairs to the nearest prototype pairs based on action similarity, followed by minimizing within-cluster distance and maximizing between-cluster distance. (c) The transition prototypes are learned from data points, capturing common ground transitions, and are further propagated to refine the dynamics.

### 3.1 MOTIVATION

The quality of the policy in MBRL significantly hinges on the accuracy of simulations generated by the world model. This model constructs a compact latent representation primarily through observation-related tasks. In the context of frame-masked reinforcement learning, which offers certain benefits, there arises a potential drawback - the risk of flattening the latent space due to padded observations. Specifically, frame masking may lead to the abandonment of some state signals from the environment, disrupting the coherence of state sequences. Additionally, frame padding could further compound this issue by mapping consecutive states to a single one. This phenomenon bears resemblance to mode collapse in Generative Adversarial Networks (GANs; Srivastava et al. (2017)) and over-smoothing in Graph Convolutional Networks (GCNs; Chen et al. (2020)).

To rejuvenate and rectify the compromised latent representation space, we propose the incorporation of the causal relationship between actions and transitions, which we refer to as "action-consistent transitions" (ACT). ACT aims to preserve the consistency of state transitions induced by similar actions. Unlike isolating the action space from the state space (Chandak et al., 2019), our central idea is to implicitly embed action information into the state representation. We hypothesize that this strategy benefits the frame-masked world model for two primary reasons: (1) From a local perspective, action sequences serve as a means to distinguish frame-padded states from their counterparts, acting as a regularization that discourages the model from overfitting to the padded observations. (2) From a global standpoint, state representation might experience occasional mismatches in the absence of full access to observations. However, with the aid of action-transition correlation, state consistency and robustness are maintained even in the absence of direct supervision.

One potential concern with this proposal is its applicability, particularly when enforcing a single transition for various states after taking identical actions. For instance, in a continuous control task,

the transition of a humanoid character may differ when it is on the ground compared to when it is in the air, despite employing the same action vector. To address this concern and ensure the generalizability of our method, we introduce a novel algorithm named SelfDreamer. SelfDreamer leverages prototypical learning and simultaneously learns two types of prototypes for actions and transitions. These prototypes are coupled into pairs, as depicted in Fig. 2a, to enforce action-transition relationships and to identify common-ground transitions for similar actions. While these dual prototypes are intertwined, we introduce two novel objective functions into Eqn. 2, $\mathcal{J}_A$ and $\mathcal{J}_T$, which will be elaborated upon in the subsequent sections.

## 3.2  ACTION-PROTOTYPE LEARNING

For prototypical learning, during each model learning iteration, we randomly sample an experience sequence of length $L$ from the replay buffer: $\{(o_t, a_t, o_{t+1})\}_{t=1}^{L} \sim \mathcal{D}$ (where reward signals are disregarded as this method is self-supervised). This process can be scaled up to form batches of sequences, contributing to a more robust distribution estimation.

To extend the application of the previously outlined mechanism from discrete action spaces to continuous ones, we initiate the procedure by randomly initializing $k$ action prototypes. Each action prototype, denoted as $c_i^A$, is an $n$-dimensional continuous vector, contingent on the action space: $c_i^A \in \mathbb{R}^n, 1 \leq i \leq k$. Drawing inspiration from the k-means clustering algorithm (MacQueen, 1967), the initial step in action-prototype learning involves clustering action data points into distinct sets, $S_i^A$, by assigning them to the nearest action prototypes (as defined in Eqn. 3). We employ the cosine distance metric (denoted as $D_C$) for measuring the distance between actions, which considers only the orientation of actions. For a more comprehensive discussion on this clustering method, please refer to Section A.2.

$$S_i^A = \{ a_p : D_C(a_p, c_i^A) \leq D_C(a_p, c_j^A) \, \forall j, \, 1 \leq j \leq k \} \, \forall i, \, 1 \leq i \leq k \tag{3}$$

Following the assignment step mentioned above, the objective function $\mathcal{J}_A$ for action-prototype learning is formulated as described in Eqn. 4. The first term in this equation aggregates the distribution of actions assigned to a specific action prototype. As depicted in the left portion of Fig. 2b, action prototypes are generated by minimizing the within-cluster cosine distances. However, to construct meaningful action prototypes, it is essential for each cluster of actions to maintain clear boundaries between one another. Consequently, the second term in the equation aims to maximize the between-cluster distance, ensuring that each cluster represents a unique domain of actions, as illustrated on the right side of Fig. 2b. Given that determining the appropriate number of action groups $k$ can be challenging even with domain knowledge, this min-max game design proves valuable. It allows for the use of a larger number of prototypes initially, and redundant ones can be subsequently dismissed in an autoregressive manner by moving them further away from those on active duty.

$$\mathcal{J}_A = \sum_{i=1}^{k} \sum_{a \in S_i^A} -D_C(c_i^A, \, a) + \sum_{i=1}^{k} \sum_{j=i+1}^{k} D_C(c_i^A, \, c_j^A) \tag{4}$$

## 3.3  TRANSITION-PROTOTYPE LEARNING

Before initiating the learning process for transition prototypes, we first input the sampled experience sequence into the current world model for RSSM state inference, as denoted by $\{s_t, a_t, s_{t+1}\}_{t=1}^{L} \forall s_t = [h_t, z_t]$. Subsequently, we define transitions $t_t$ as the residuals between adjacent deterministic states, specifically $h_{t+1} - h_t$, excluding the stochastic states to ensure a variance-free representation. Lastly, since we couple actions and transitions for both the prototypes and the data points, we can then partition transitions into $k$ transition sets $S_i^T$ corresponding to the transition prototypes initialized from Gaussian distribution: $c_i^T \in \mathbb{R}^m, \forall 1 \leq i \leq k$, where $m$ represents the dimension of the deterministic states.

$$S_i^T = \{ t_p : a_p \in S_i^A \} \, \forall i, \, 1 \leq i \leq k \tag{5}$$

As visualized in Fig. 2c, a mutual information exchange occurs between the prototypes and the transition data points, where a transition prototype learns from the transitions and subsequently propagates the integrated transition information back into the system. This interaction is formalized through the objective function $\mathcal{J}_T$ (Eqn. 6), which employs gradient stopping ($sg$) to prevent backpropagation and fix certain models or prototypes. The first term of this objective function trains each transition prototype to fit all transition data points, weighted by the cosine similarity of their associated actions. This results in the learning of a shared latent encoding for similar actions, even when these actions are applied within different states. Furthermore, the weightings range from $-1$ to 1, so negative cosine similarity encourages contrastive learning among transition prototypes, preventing the representation from collapsing. The latter term of the objective function is the key to this algorithm, guiding the world model by minimizing the cosine distance between each transition and its corresponding transition prototype. Since transitions are derived from the feedforward process of the world model, gradients can flow through the sequence of inferred model states, thereby enforcing the action-transition regularization discussed in Section 3.1.

$$\mathcal{J}_T = \sum_{i=1}^{k} \sum_{a,t} -sg(S_C(c_i^A, a)) * D_C(c_i^T, \ sg(t)) + \sum_{i=1}^{k} \sum_{t \in S_i^T} -D_C(sg(c_i^T), \ t) \tag{6}$$

## 4 EXPERIMENTS

**Environment Setup.**  In our evaluation, we concentrate primarily on image-based RL, and as a result, we assess the performance of our method and the baseline algorithms within the context of the DeepMind Control Suite (DMC; Tassa et al. (2018)). This suite encompasses a diverse array of continuous control tasks, and we have selected six of these tasks for our evaluation, consistent with the settings employed in prior works (Deng et al., 2022). The selected tasks include *Cartpole Swingup Sparse*, *Cheetah Run*, *Cup Catch*, *Finger Spin*, *Reacher Easy*, and *Walker Run*. To ensure a comprehensive assessment, we consider three distinct environment setups: (1) Standard DMC: In this setup, we adhere to the default configuration of the DMC, which serves as our baseline environment. (2) Frame-masked DMC: This setup aligns with the frame-masking paradigm described in Algorithm 1 of our paper. Within this configuration, observations in the replay buffer are subjected to frame masking, where observations are masked and padded at a frame-masking period denoted as $P$. In our experiments, we employ frame-masking periods of 2 and 3 to simulate the data-efficient framework proposed in our paper. (3) Natural background DMC: In line with the approach introduced by Nguyen et al. (2021), we introduce a setup where the background in the DMC environment is replaced with random natural videos. In this configuration, the task Cartpole Swingup Sparse is substituted with Cartpole Swingup. Further details on this setup can be found in the original paper.

**Baselines.**  In this study, we focus on MBRL, and our method is compared with several state-of-the-art MBRL frameworks. Specifically, we include Dreamer (Hafner et al., 2021), a foundational algorithm that has paved the way for subsequent research (Chen et al., 2022; Seo et al., 2023; Wu et al., 2023). Additionally, we evaluate our method against two reconstruction-free variations: TPC (Nguyen et al., 2021), which leverages contrastive predictive coding (Oord et al., 2018), and DreamerPro (Deng et al., 2022), a strong baseline that has demonstrated superior or comparable performance to Dreamer, TPC, and Dreaming (Okada & Taniguchi, 2021), another MBRL baseline.

**Evaluation Protocol.**  In accordance with the evaluation protocol established by Deng et al. (2022), our evaluation procedure adheres to the following guidelines: For each of the selected tasks, every model undergoes training for a duration equivalent to $1M$ environment steps. This corresponds to $500K$ actor steps, as the action repeat is configured to two. To assess the performance, the evaluation return is computed at intervals of $10K$ training steps, and the results are averaged over ten episodes for each evaluation point.

### 4.1 PERFORMANCE IN STANDARD DMC

In our initial set of experiments, we sought to assess the generality and performance of SelfDreamer by comparing it to the baseline methods within the standard DMC. The results of this comparison are

Table 1: Final performance in standard DMC.

| Task | Dreamer | TPC | DreamerPro | SelfDreamer |
|---|---|---|---|---|
| Cartpole Swingup Sparse | $810 \pm 39$ | $811 \pm 20$ | $792 \pm 29$ | $\mathbf{837 \pm 5}$ |
| Cheetah Run | $755 \pm 208$ | $713 \pm 37$ | $892 \pm 11$ | $\mathbf{901 \pm 11}$ |
| Cup Catch | $679 \pm 410$ | $926 \pm 26$ | $957 \pm 7$ | $\mathbf{958 \pm 3}$ |
| Finger Spin | $553 \pm 305$ | $663 \pm 218$ | $527 \pm 78$ | $\mathbf{744 \pm 138}$ |
| Reacher Easy | $849 \pm 75$ | $487 \pm 112$ | $930 \pm 30$ | $\mathbf{962 \pm 12}$ |
| Walker Run | $649 \pm 162$ | $175 \pm 16$ | $620 \pm 76$ | $\mathbf{778 \pm 18}$ |

presented in Table 1. The results reveal that SelfDreamer consistently outperforms all three baseline methods across the evaluated tasks. Specifically, when compared to Dreamer, TPC, and DreamerPro, SelfDreamer exhibits performance improvements of 22%, 75%, and 12%, respectively, on average across the tasks. Additionally, it is noteworthy that SelfDreamer demonstrates relatively minor standard deviations in performance, except for Finger Spin, which exhibits relatively higher instability across all methods. A particular highlight is the performance on Walker Run, where SelfDreamer achieves a remarkable 25% improvement compared to DreamerPro. These findings underscore the effectiveness of the proposed heuristic employed by SelfDreamer, namely, action-consistent transitions, in enhancing the performance of model-based reinforcement learning for continuous control tasks. The results indicate that SelfDreamer holds promise as a robust and competitive approach within the standard DMC setting.

## 4.2 PERFORMANCE IN FRAME-MASKED DMC

Table 2: Final performance in frame-masked DMC (**2x** less state signals).

| Task | Dreamer | TPC | DreamerPro | SelfDreamer |
|---|---|---|---|---|
| Cartpole Swingup Sparse | $831 \pm 14$ | $831 \pm 7$ | $788 \pm 30$ | $\mathbf{837 \pm 2}$ |
| Cheetah Run | $875 \pm 18$ | $787 \pm 66$ | $795 \pm 111$ | $\mathbf{881 \pm 16}$ |
| Cup Catch | $724 \pm 322$ | $950 \pm 10$ | $959 \pm 12$ | $\mathbf{960 \pm 4}$ |
| Finger Spin | $646 \pm 199$ | $939 \pm 21$ | $973 \pm 7$ | $\mathbf{977 \pm 2}$ |
| Reacher Easy | $845 \pm 78$ | $331 \pm 65$ | $969 \pm 7$ | $\mathbf{969 \pm 2}$ |
| Walker Run | $264 \pm 94$ | $132 \pm 49$ | $616 \pm 73$ | $\mathbf{698 \pm 17}$ |

Table 3: Final performance in frame-masked DMC (**3x** less state signals).

| Task | Dreamer | TPC | DreamerPro | SelfDreamer |
|---|---|---|---|---|
| Cartpole Swingup Sparse | $818 \pm 29$ | $742 \pm 74$ | $767 \pm 24$ | $\mathbf{822 \pm 19}$ |
| Cheetah Run | $806 \pm 103$ | $717 \pm 94$ | $803 \pm 54$ | $\mathbf{858 \pm 4}$ |
| Cup Catch | $946 \pm 11$ | $939 \pm 5$ | $954 \pm 6$ | $\mathbf{954 \pm 4}$ |
| Finger Spin | $771 \pm 189$ | $584 \pm 39$ | $683 \pm 190$ | $\mathbf{813 \pm 110}$ |
| Reacher Easy | $815 \pm 49$ | $380 \pm 24$ | $942 \pm 39$ | $\mathbf{956 \pm 13}$ |
| Walker Run | $100 \pm 54$ | $87 \pm 27$ | $406 \pm 9$ | $\mathbf{471 \pm 24}$ |

In our second set of experiments, we delve into the realm of frame-masked reinforcement learning, as introduced in Section 2.1. We present the results obtained when employing a frame-masking period set to 2 (Table 2) and another set to 3 (Table 3) to investigate the impact on model performance when reducing visual information.

Table 2 presents the results when the frame-masking period is set to 2. Notably, we observe that for Dreamer and TPC, there is a drop in performance by 3% and 7%, respectively, while DreamerPro demonstrates an improvement of 15%. This improvement extends to certain tasks, with Cartpole Swingup Sparse, Cheetah Run, Cup Catch, and Finger Spin experiencing performance gains of 2%, 4%, 3%, and 25%, respectively. These results underscore the potential of frame-masked reinforcement learning to achieve higher data efficiency while maintaining or even enhancing performance.

Moreover, when SelfDreamer is applied, the final return is further improved by $18\%$ compared to DreamerPro "without" frame masking, achieving double data efficiency and highlighting the benefits of the action-transition dual prototypes introduced in Section 3.

Table 3 explores the impact of setting the frame-masking period to 3. In this scenario, Dreamer, TPC, and DreamerPro all experience diminished final returns due to the loss of visual information, with reductions of $4\%$, $13\%$, and $11\%$, respectively. This impact is particularly pronounced in the challenging Walker Run task, where the performance degradation is notable, especially when compared to the results from Table 2. Despite these challenges, SelfDreamer continues to outperform the best baseline, DreamerPro, by $8\%$ on average across all tasks. This suggests that SelfDreamer effectively assigns a robust latent state in the world model, aiding downstream behavior learning, even in scenarios with reduced visual information.

### 4.3 Performance in natural background DMC

Table 4: Final performance in natural background DMC.

| Task | Dreamer | TPC | DreamerPro | SelfDreamer |
|---|---|---|---|---|
| Cartpole Swingup | $123 \pm 26$ | $567 \pm 60$ | $636 \pm 95$ | $\mathbf{731 \pm 51}$ |
| Cheetah Run | $26 \pm 8$ | $349 \pm 53$ | $356 \pm 15$ | $\mathbf{404 \pm 21}$ |
| Cup Catch | $57 \pm 51$ | $536 \pm 93$ | $555 \pm 91$ | $\mathbf{661 \pm 29}$ |
| Finger Spin | $2 \pm 2$ | $309 \pm 24$ | $801 \pm 233$ | $\mathbf{916 \pm 38}$ |
| Reacher Easy | $101 \pm 47$ | $\mathbf{705 \pm 97}$ | $672 \pm 168$ | $701 \pm 23$ |
| Walker Run | $39 \pm 1$ | $149 \pm 11$ | $383 \pm 41$ | $\mathbf{409 \pm 11}$ |

In our final set of experiments, we explore the performance of SelfDreamer in the context of natural background DMC, where nuisance and task-irrelevant information are introduced to distract the learning process. Table 4 presents the results of these experiments, with a focus on the comparison between SelfDreamer and DreamerPro, which serves as the foundation for our method. The results indicate that SelfDreamer exhibits a $12\%$ performance improvement on average across all tasks when compared to DreamerPro. Additionally, SelfDreamer demonstrates more stable final performance across the tasks. These findings suggest that SelfDreamer is capable of generalizing to model-based reinforcement learning scenarios with noisy observations and distractions introduced by task-irrelevant information. Furthermore, the results imply that SelfDreamer effectively prioritizes the world model's learning of task-relevant information even in challenging and distracting environments, showcasing its robustness and adaptability.

## 5 Conclusion and future direction

In this paper, we introduce a novel reinforcement learning framework called "frame-masked RL," which effectively learns from sparse state signals, thereby achieving higher data efficiency. Furthermore, we have presented "SelfDreamer," a model-based algorithm that leverages prototypical learning and action-transition dual prototypes to mitigate representation flattening issues in the frame-masked world model. Our empirical results, based on continuous control tasks within the DeepMind Control Suite, demonstrate that SelfDreamer consistently outperforms three state-of-the-art methods across frame-masked DMC and other experimental settings, highlighting its versatility and effectiveness in model-based reinforcement learning.

As our current focus primarily centers on continuous control tasks and model-based RL methods, future research avenues include extending the application of frame masking and investigating the heuristic of action-consistent transitions for (1) tasks involving discrete actions and a diverse array of states, such as those found in the Atari benchmark (Bellemare et al., 2013), and (2) model-free RL methods (Yarats et al., 2022). Additionally, addressing distribution mismatch concerns is an important consideration. In this work, the testing policy is constrained to utilize frame-masked sequences of states. Future research could explore methods to enable the evaluation process to leverage full observations, thereby paving the way for further advancements in this direction. Our work serves as a foundational stepping stone for these prospective research endeavors.

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
