# OpenReview forum: "SelfDreamer: Dual-Prototypical Regularization for Frame-masked Model-based Reinforcement Learning"
_ICLR.cc/2024/Conference — ICLR 2024 Conference Withdrawn Submission_

### Official Review · Reviewer_Ngz6 · 2023-10-20

**Soundness:** 2 fair
**Presentation:** 2 fair
**Contribution:** 2 fair
**Rating:** 3
**Confidence:** 5

**Summary:**

This paper studies a new RL setting, namely frame-masked RL, where observation frames are only available at intervals of certain time steps. This setting is claimed to be valuable in the context of edge-cloud co-design, where skipped observation frames reduce data transmission overhead. However, naively padding with the last available frame may cause representation collapse. This paper proposes SelfDreamer, where two kinds of prototypes, namely action prototypes and transition prototypes, are learned, and clusters in the action space are distilled into the representation transition space to prevent potential collapse. SelfDreamer is built upon Dreamer and demonstrates improved performance in standard, frame-masked, and natural background DMC.

**Strengths:**

1. An interesting new setting of visual RL is proposed.
2. Representation collapse is recognized as the major difficulty of this setting, and cluster distillation with learned prototypes is introduced to prevent collapse.
3. Experimental results on several variants of DMC demonstrate the superiority of SelfDreamer.

**Weaknesses:**

1. A sufficient and clear discussion on the difference with or advantage upon frame skipping (action repeating) is missed. Some important baselines are also not compared. See the question 2 below.
2. There is no analysis of whether the proposed SelfDreamer actually mitigates representation collapse. Also, see question 3 below.
3. Poor writing. There are several sentences that are hard to read for the first time, particularly for the method section. For example:
   - 'Even attempts to mitigate this issue, such as padding missing frames with the latest valid frames': why can padding missing frames mitigate the distributional disparity problem?
   - 'particularly when enforcing a single transition for various states after taking identical actions': it is hard to understand what it means. But I can guess the meaning from the following example of continuous control.
   - 'To extend the application of the previously outlined mechanism from discrete action spaces': where is the 'previously outlined mechanism'?
   - 'Addressing distribution mismatch concerns is an important consideration': how are the distribution mismatch concerns connected to this paper?
   - etc.

Also, Figure 2 is not informative, and I recommend recalling the overall objective in the last of the method section.

**Questions:**

1. Does the edge-cloud co-design predicament actually exist? Can you provide some examples of real-world applications utilizing RL in the edge-cloud context? If so, does the frame-masked paradigm provide an accurate and reasonable abstract of this context?
2. What is the essential difference or advantage between frame-masked RL and frame skipping (action repeating)?
   - In Algorithem 1, actions are sampled $a_t \sim \pi_\phi\left(a_t \mid o_t\right)$ after that observations are replaced as $o_t \leftarrow o_{t-1}$ However, in the main text, 'the true observation is discarded after the reward function calculation'. So, which observation indeed is utilized to determine the action? I guess the text is correct, and there is a typo in the algorithm.
   - Some trivial baselines should be compared: (1) accumulating the rewards in the frame-masking period (P time steps) for data collection and repeating the same action P times for policy execution. Thus, the frame-masked RL is translated to a standard RL with a lower frequency. (2) accumulating the rewards and constructing a larger space $\mathcal{A}^P$ of successive P actions. Since P is set to 2 and 3 in this paper, I think these baselines are both feasible to prevent representation collapse and should be compared.
3. Since the recurrent model $h_t=f_\theta\left(h_{t-1}, z_{t-1}, a_{t-1}\right)$ has already embed actions in the transitions, why should we explicitly utilize a distillation of clustering from action space into transition space (so-called "action-consistent transitions" in the main text)? I acknowledge the function of these auxiliary losses as a stronger regularization. Can the author provide more insights or, more essentially, empirical evidence?

---

### Official Review · Reviewer_Euds · 2023-10-20

**Soundness:** 2 fair
**Presentation:** 3 good
**Contribution:** 2 fair
**Rating:** 5
**Confidence:** 4

**Summary:**

Considering data transmission between edge devices and cloud servers, this work proposes frame-masked RL. Based on Dreamer-based world models, the proposed method, SelfDreamer, leverages action-transition dual prototypes and transition-transition dual prototypes. Experiments in DMC, including standard setting, frame-masked setting, natural background setting, show the effectiveness of SelfDreamer.

**Strengths:**

- This work considers frame-masked RL, which considers the data collection in edge devices and cloud servers.

- The proposed two terms seem novel and helpful for world models.

- Experiments on DMC show that SelfDreamer significantly outperforms than baselines in the standard setting and natural background setting.

**Weaknesses:**

- This work proposes a novel paradigm termed frame-masked RL. However, it is better to provide a formal definition of it.

- The proposed frame-masked RL is really close to frame skipping, which is a standard training trick in Atari and DMC. Although this paper has mentioned this (Lines 19-22 on Page 4), I hope to see more discussion about them, especially their similarities and differences. For example, in my opinion, frame skipping will take the same action across all these frames but frame-masked RL can take different actions, moreover, frame skipping will only add the first frame to the replay buffer for training while frame-masked RL will add all the frames into the replay buffer, are these right? I think a thorough discussion will better position this work.

- Due to the lack of a definition for it, it is difficult to intuitively understand why the proposed algorithm would be helpful for frame-masked RL.

- The chosen tasks do not show the challenge of frame-masked RL. Comparing Table 1-3, Dreamer performs similarly in 5 tasks and only performs poorly in Walker-run. In my opinion, one of the reasons might be the action repeat, PlaNet chooses action repeat as cartpole (R = 8), reacher (R = 4), cheetah (R = 4), finger (R = 2), cup (R = 4), walker (R = 2). As Dreamer take the action repeat for all tasks as 2, the performance of Dreamer will not decrease when we take action repeat to 4 or 6 (i.e., frame-masked rate=2 or 3). In Table 12 of Dreamer, Walker-run with action repeat 4 is significantly lower than action repeat 2, thus Walker-run is challenging under frame-masked RL. Consequently, it will be more reasonable to choose more challenging environments, which do not need action repeat, for designing algorithms for frame-masked RL.

- The proposed method, SelfDreamer, does not significantly outperform baselines in the setting of frame-masked RL, especially in 4x and 5x experiments (Table 6-7).

- Lack of related works about word models, including representation learning of word models and improving generalization for world models[1-7].

Overall, I vote for boardline rejection currently since the discussion between frame-masked RL and frame skipping is inadequate, and the performance of the proposed method in frame-masked RL is not such surprising. As I still think the proposed terms can benefit Dreamer, I will keep active in the following discussion and accordingly update my score if the authors can address my concerns.

[1] Learning Task Informed Abstractions

[2] DreamingV2: Reinforcement Learning with Discrete World Models without Reconstruction

[3] Denoised MDPs: Learning World Models Better Than the World Itself

[4] Task Aware Dreamer for Task Generalization in Reinforcement Learning

[5] Simplified Temporal Consistency Reinforcement Learning

[6] Mastering Diverse Domains through World Models

[7] Planning to Explore via Self-Supervised World Models

**Questions:**

- As Dreamer will always choose action repeat = 2, do all algorithms in experiments take action repeat=2? In other words, 2x less state signals in Table 2 means that we take action every 4 frames (4=2*2).

- What is the main training objective of SelfDreamer? Will it use the origin terms in Dreamer($J_{KL},J_R,J_O$)?

- Typos:

Line 5 in Page 4:  ”frame-masked RL,” should be "frame-masked RL",

Line 22 in Page 6: Section A.2 should be Appendix A.2

---

### Official Review · Reviewer_GHX9 · 2023-10-30

**Soundness:** 2 fair
**Presentation:** 1 poor
**Contribution:** 2 fair
**Rating:** 3
**Confidence:** 4

**Summary:**

The paper introduces a modification of the Dreamer algorithm for model-based reinforcement learning called “SelfDreamer”, specifically meant to address the issue of learning from “frame-masked” observations. Frame-masked observations occur when, for POMDP trajectories, only one in N consecutive observations is observed while the others are “masked”, usually set equal to the last non-masked observation. This problem arises in practice when learning from data which is collected by edge devices (e.g. drones, robot arms) and then processed by a centralized server in settings with data transmission bottlenecks, as high dimensional video observations would need to be transferred over the network.

The paper first recaps the Dreamer architecture, then introduces “SelfDreamer”, with the inclusion of action and transition prototypes, each learned with specific loss functions.

Then, several experiments are conducted in the Deepmind Control Suite (DMC), comparing SelfDreamer with Dreamer, TMC and DreamerPro as baselines.

**Strengths:**

The paper proposes an interesting modification to the Dreamer method for model-based RL. Overall, the idea of applying some kind of regularization over recurrent model hidden state transitions appears to be a good idea, if we are concerned about the accuracy of Dreamer simulations.

The experimental results seem to consistently show the SelfDreamer technique to beat all other baselines for all experiments, and moreover they seem to show that the technique leads to lower variances over performance scores.

**Weaknesses:**

Overall, the way that the technique is presented is confusing, and does not adequately convey the rationale behind the specific choice of mechanism for action/transition prototype learning. To the best of my understanding, thanks to this action/transition clustering, the model is forced to learn an RNN latent space that tends to change according to a limited set of transition prototypes, and specifically undergoes different transitions when different actions are used, irrespectively of whether the observation is the same due to frame masking. Transitions in latent space (the difference of hidden representations over one RNN timestep) are clustered by prototypes, each paired with an action cluster prototype. At each timestep, the current transition is pushed close to the transition prototype paired with the current action cluster’s prototype. At the same time the transition prototypes themselves are trained by pushing them close to all RNN transitions data points, but weighted by negative cosine distance of the paired action to the action prototype. It would be interesting to have spent more time explaining the reasoning behind choosing this precise way of designing the method for prototype learning.

More importantly, the experimental results do not seem to support the central thesis of the paper, namely that the SelfDreamer technique is beneficial specifically in the frame-masked setting:
* SelfDreamer performs better than the alternatives from the very beginning (without frame-masking).
* In the experiment with 2x frame masking, none of the methods (both SelfDreamer and baselines) seem to significantly degrade in performance. (With the exception of the WalkerRun task, for which Dreamer and TMC degrade in performance. In any case, DreamerPro does not degrade in performance.)
* With 3x masking, we have a small drop in performance across the board for all methods, including SelfDreamer. (Other methods have confidence intervals over performance too big to precisely quantify the drop in performance and compare it to SelfDreamer. Moreover, the drop in performance for Walker Run behaves similarly to the 2x masking setting, in which performance drops greatly for Dreamer and TMC, but not as much for DreamerPro. SelfDreamer beats DreamerPro, as it beat DreamerPro by a similar margin even before any frame masking)
* The 4x and 5x masking results from the appendix further reinforce the above point, that SelfDreamer does not seem to offer a significant relative resilience to masking compared to DreamerPro.

Overall, it seems that SelfDreamer is indeed a useful modification to Dreamer. What it seems not to be is a method that should be viewed specifically from the lens of improving performance when doing frame-masking. It seems that the key insight that improves performance in the frame-masking a setting is that of using prototypes in general, already introduced with DreamerPro. SelfDreamer seems then a complex technique that delivers a small improvement over DreamerPro for all scenarios.

As an additional point: while the paper spends time introducing the setting of edge-cloud dreamer training, the only experiment performed is on Deepmind Control Suite. It seems that, if such an edge-cloud setting were to be the main topic of the paper, a real world experiment in such a setting should have been included.

**Questions:**

I have the following questions:
* Is my above description of the intuition behind the SelfDreamer prototype training method correct? If not, why? Can the rationale behind the method be more clearly and explicitly laid out?
* Can you address the point, in “Weaknesses”, about how the SelfDreamer approach seems not to be relatively more useful than DreamerPro for the frame-masking setting? (it seems to be better in general for all settings in absolute terms, but not "relatively" better in the frame-masking setting)
* It is unclear whether SelfDreamer is built on Dreamer, simply adding the action and transition prototype modifications, or if it is built on DreamerPro, which already employs state prototypes. In page 4, it is written that “DreamerPro [...] serves as the basis of our work”, does this mean that you build directly on top of it (your method uses the DreamerPro prototype method for states, with no reconstruction), or does it mean that you have been inspired by its approach to propose your own action/transition prototype method, but this method is implemented on top of the original dreamer? What is the full loss optimized by your method? Does this loss have a reconstruction term, or a DreamerPro prototype term?
* Depending on the answer to the above question: If your method is built directly on top of the DreamerPro technique, can you conduct an ablation experiment, consisting of applying your action/transition prototyping technique on top of standard Dreamer? I imagine that such an experiment would help show whether the DreamerPro prototypes are the actual "key" ingredient that make the method work in the frame-masking setting, potentially invalidating some arguments laid out in this paper.
* What is the number of prototypes for DreamerPro in the experiments? I cannot seem to find this number in the hyperparameters in the appendix. Again, depending on the answer to question 3: If the SelfDreamer method does NOT build directly on top of the DreamerPro implementation, could it be that the difference in performance between SelfDreamer and DreamerPro is due to the number of prototypes that are used?

---

### Official Review · Reviewer_CGK5 · 2023-11-01

**Soundness:** 3 good
**Presentation:** 3 good
**Contribution:** 2 fair
**Rating:** 5
**Confidence:** 5

**Summary:**

The paper introduces a novel approach called "frame-masked RL," designed to enhance data efficiency in reinforcement learning scenarios where high-dimensional state representations pose challenges in data transmission. It proposes a model-based algorithm, "SelfDreamer," which employs action-transition dual prototypes to mitigate information loss due to frame masking. The method is evaluated on DeepMind Control Suite tasks, demonstrating superior or comparable performance while utilizing only half of the observations from the environment.

**Strengths:**

1. This paper is well-written and easy to follow. The paper effectively communicates complex concepts, such as frame masking and the SelfDreamer algorithm, using clear language and diagrams. This clarity aids in understanding the proposed methodology. The clarity in the presentation enhances the quality of understanding for readers. The method's reliance on self-supervised regularization, particularly in the context of frame-masked RL, highlights the method's quality and innovative nature.

2. The paper introduces a novel paradigm, "frame-masked RL," addressing the challenges of high-dimensional state representations in reinforcement learning, showcasing originality in problem formulation. SelfDreamer's integration of action-transition dual prototypes represents a creative combination of existing ideas, demonstrating originality in the methodology.

**Weaknesses:**

1. The paper lacks a comprehensive comparison and discussion of existing methods. There are tons of model-free efficient RL works recently (state augmentation [1, 2] and masked auxiliary losses [3, 4, 5]). I think it is worth discussing the recent progress of efficient RL in the related work session or adding baselines for comparison.

[1] Laskin, M., Lee, K., Stooke, A., Pinto, L., Abbeel, P., & Srinivas, A. (2020). Reinforcement learning with augmented data. Advances in neural information processing systems, 33, 19884-19895.

[2] Yarats, D., Fergus, R., Lazaric, A., & Pinto, L. (2021). Mastering visual continuous control: Improved data-augmented reinforcement learning. arXiv preprint arXiv:2107.09645.

[3] Schwarzer, M., Anand, A., Goel, R., Hjelm, R. D., Courville, A., & Bachman, P. (2020). Data-efficient reinforcement learning with self-predictive representations. arXiv preprint arXiv:2007.05929.

[4] He, T., Zhang, Y., Ren, K., Liu, M., Wang, C., Zhang, W., ... & Li, D. (2022). Reinforcement learning with automated auxiliary loss search. Advances in Neural Information Processing Systems, 35, 1820-1834.

[5] Zhu, J., Xia, Y., Wu, L., Deng, J., Zhou, W., Qin, T., ... & Li, H. (2022). Masked contrastive representation learning for reinforcement learning. IEEE Transactions on Pattern Analysis and Machine Intelligence, 45(3), 3421-3433.

2. The evaluation focuses on a limited set of tasks from the DeepMind Control Suite. The paper could benefit from a broader evaluation across diverse RL benchmarks to demonstrate the method's applicability and generalizability. Also, there are a lot of other environments in DMC. For a work claiming superior sample efficiency, I think it needs more empirical experimental results. It is dangerous to conclude sample efficiency only from limited environments in one experimental domain.

**Questions:**

1. Could you provide a more extensive comparison with existing model-based RL methods? Detailed comparative experiments and analysis would strengthen the paper's contributions.

2. How do you anticipate the proposed SelfDreamer approach performing tasks beyond the DeepMind Control Suite? Are there specific challenges or limitations that might arise in different environments or domains?

3. I think there are a lot of design choices for the frame-masking training. Could the authors provide a more in-depth ablation study on it (e.g., Frame masking horizon...) ?

I'd happily raise my evaluation scores if my concerns are addressed. Thanks!